# Study of Solid Particle Erosion on Helicopter Rotor Blades Surfaces

**Xupeng Bai [1], Yongming Yao [1,*], Zhiwu Han [2], Junqiu Zhang [2] and Shuaijun Zhang [2]**

[1]  School of Mechanical and Aerospace Engineering, Jilin University, Changchun 130022, China; baixp18@mails.jlu.edu.cn

[2]  Key Laboratory of Bionic Engineering (Ministry of Education), Jilin University, Changchun 130012, China; zwhan@jlu.edu.cn (Z.H.); junqiuzhang@jlu.edu.cn (J.Z.); sjzhang18@mails.jlu.edu.cn (S.Z.)

\*  Correspondence: ymyao@jlu.edu.cn; Tel.: +86-158-4401-0057

**Abstract:** In this study, titanium alloy (Ti-4Al-1.5Mn), magnesium alloy (Mg-Li9-A3-Zn3), or aluminum alloy (Al7075-T6) were used to construct the shell model of helicopter rotor blade to study the solid particle erosion of helicopter rotor blades. The erosion resistance of the three materials at different angles of attack (6°, 3°, or 0°) and particle collision speeds (70, 150, or 220 m/s) was examined using the finite volume method, the discrete phase method, and erosion models. In addition, the leading edge of the helicopter blades was coated with two types of bionic anti-erosion coating layers (V- and VC-type), in an attempt to improve erosion resistance at the angles of attack and particle collision speeds given above. The results showed that Ti-4Al-1.5Mn had the best erosion resistance at high speed, followed by Al7075-T6 and Mg-Li9-A3-Zn3. The angle of attack appeared to affect only the surface area of the blade erosion, while the erosion rate was not affected. Finally, the results of this article showed that the V-type bionic coating had better erosion resistance than the VC-type coating at the same impact speeds. The angle of attack did not have a significant effect on the erosion rate of the bionic coating.

**Keywords:** numerical study; solid particle erosion; materials; bionic coating

## 1. Introduction

Solid particle erosion (SPE) generally refers to erosion and wear on a material surface caused by particle flow or dust at a certain speed and scale. SPE depends on the particle velocity and impact angle, in addition to many other factors [1]. Due to the specialized nature of the work environment, where helicopters and transport aircrafts often take off and land in a desert or on the battlefield, dust erosion is frequently a problem [2]. Erosion caused by high-speed solids can result in component loss, especially in the main rotor blades of helicopters and the turbine blades of aircrafts [3]. The ability to predict the erosion of helicopter rotors blades is extremely important for optimal and safe operation.

There have been numerous studies of flow erosion on blades and complex-shaped surfaces. Shin et al. carried out a numerical analysis using Fluent software (Ansys Inc., Canonsburg, PA, USA) of SPE on the blade surfaces of a helicopter main rotor rotating at high rotation speed in a dense, solid particle airflow at low altitude [4]. The authors' results showed that, on the surface, the impact velocity of solid particles increases linearly to the tip of the rotor blade, where the erosion rate distribution is higher. Shin et al. also reported that the circumferential velocity, angle of attack, and blade length were related to SPE on the blade surface of a rotating rotor. Other detailed studies are available in the literature [5–9]. Early numerical studies on the SPE of rotating blades determined the relevant factors and theoretical equations that affect the SPE of rotating blades, which provided ideas for later researchers, but the content of the study did not consider the material properties of the blades themselves. However, on

the basis of previous researches, this article considers both the impact of particle velocity and other factors on blade erosion, as well as the impact of blade material properties on blade erosion.

In numerical studies of solid mechanics, the finite element method (FEM), smooth particle hydrodynamics, and micro-scale dynamic models are commonly applied [10–15]. Liu et al. used commercial software (Abaqus; ABAQUS Inc., Providence, RI, USA) and FEM to conduct three-dimensional (3D) numerical erosion research; the authors studied the effects of the impact velocity and impact angle of particles with spherical and sharp-edge geometries on erosion, with good consistency seen between numerical and experimental results [16]. The research method of Liu et al. has a great inspiration for this article, but in practice, the shape of the particles is uncontrollable and cannot be used as the main erosion factor.

Much of the flow erosion research performed to date has focused on engines, due to the associated fuel costs. As the fan components have a decisive influence on engine thrust and fuel consumption, research on the SPE of the fan blade's leading edge is of great significance. Hergt et al. pointed out that the shape of the leading fan blade edge has an important influence on aerodynamic performance in the case of transonic flow [17]. However, Hergt et al. did not conduct further research on the effect of blade leading edge shape on erosion. Fewer studies on SPE of helicopter rotor blades have appeared. Edwards et al. [18] (1998) conducted early computational fluid dynamics (CFD) studies aimed at predicting erosion. In recent years, new methods such as the discrete element method in solid mechanics, and CFD in fluid mechanics, have been applied increasingly to models of major geometric erosion [19,20]. Parsi et al. performed a numerical erosion analysis with ANSYS Fluent to determine the erosion of solid particles in the pipeline [21]. However, the authors did not discuss and study the erosion protection at the elbow. The resulting erosion equations have since been validated by other experts and scholars [22–24].

Bionics has been applied in the field of aviation since the 1990s. Peery et al. simulated the various body postures of fish swimming in water in response to the incoming flow conditions, and proposed the concept of adaptive flexible deformed wings to improve aircraft aerodynamic performance [25]. Miklosovic et al. [26], Pedro et al. [27], and Johari et al. [28] all studied the fin wings of bionic humpback whales, these researchers are very innovative in improving the aerodynamic characteristics of the blades through the bionics perspective, but the researchers have not considered the erosion of the bionic blades. Guangyu et al. [29] carried out extensive research on helicopter coatings, where such coatings can improve the erosion resistance of compressor blades [30]. Lee [31] assumed that the coating was a linear elastomer, and calculated the residual and thermal stresses therein using the boundary element method; their results showed that stress in the corners and free edges causes cracking and peeling of the coating, leading directly to coating failure. Pan et al. [32] studied the combination of particles and substrates at different temperatures to determine the optimal coating method. Many studies have reported that temperature shock does more damage to the coating than a constant high temperature in coating aging experiments, and mechanical factors have also been considered in the simulation experiments of coating failure [33,34]. Installing coatings is a popular anti-erosion method today. It is important to study whether it is feasible to install bionic structures as coatings on blades to prevent erosion.

To reduce the operating costs of helicopters, alloy materials are commonly used to cover the helicopter rotor blades. In this study, the following models were used to test the SPE of helicopter rotor blades in which the outer skin of the helicopter blades was constructed from titanium alloy (Ti-4Al-1.5Mn), magnesium alloy (Mg-Li9-A3-Zn3,) and aluminum alloy (Al7075-T6). In these models, the leading edge of the blade is fitted with a V- or VC-type bionic anti-erosion coating. The alloy materials, with and without bionic coatings, were then tested for erosion resistance with respect to changes in the particle speed and angle of attack, in an attempt to optimize the anti-erosion properties of helicopter rotor blades.

## 2. Erosion Equations and CFD Setup

### 2.1. Erosion Equations

The erosion of helicopter blade models was examined numerically using ANSYS 15.0 Fluent (Ansys Inc.). The Euler–Lagrangian method, fluid flow equation, and particle motion equation were examined using finite volume and discrete phase methods.

To calculate changes in pressure, velocity, and kinetic energy in simulated models, conservation equations for mass and momentum are required. The Reynolds transformation form of the Navier–Stokes equation is necessary to solve the momentum and energy conservation equations. The generalized equation of state for the Reynolds-average Navier–Stokes equation is given by Equations (1) and (2):

$$\frac{\partial \rho}{\partial t} + \frac{\partial}{\partial x_i}(\rho U_i) = 0 \tag{1}$$

$$\frac{\partial}{\partial t}(\rho U_i) + \frac{\partial}{\partial x_i}(\rho U_i U_j) = -\frac{\partial P}{\partial x_i} + \frac{\partial}{\partial x_j}\left[\mu\left(\frac{\partial U_i}{\partial x_j} + \frac{\partial U_j}{\partial x_i} - \frac{2}{3}\delta_{ij}\frac{\partial U_k}{\partial x_k}\right)\right] + \frac{\partial}{\partial x_i}(\overline{-\rho u'_k u'_j}) \tag{2}$$

Equations (3) and (4) describe the $k$-$\omega$ SST turbulence model in CFD erosion analysis, In the $k$-$\omega$ SST turbulence model, there are two important parameters: turbulent kinetic energy $k$ and turbulent vortex frequency $\omega$:

$$\frac{\partial(\rho k)}{\partial t} + \frac{\partial(\rho U_i k)}{\partial x_i} = \overline{P_k} - \beta^* \rho k \omega + \frac{\partial}{\partial x_i}\left[(\mu + \sigma_k \mu_t)\frac{\partial k}{\partial x_i}\right] \tag{3}$$

$$\begin{aligned}\frac{\partial(\rho \omega)}{\partial t} + \frac{\partial(\rho U_i \omega)}{\partial x_i} &= \alpha \rho S^2 - \beta \rho \omega^2 + \frac{\partial}{\partial x_i}\left[(\mu + \sigma_\omega \mu_t)\frac{\partial \omega}{\partial x_i}\right] \\ &\quad + 2(1 - F_1)\rho \sigma_{\omega 2}\frac{1}{\omega}\frac{\partial k}{\partial x_i}\frac{\partial \omega}{\partial x_i}\end{aligned} \tag{4}$$

where $\rho$ is the fluid density, $\mu_t$ is the turbulent viscosity, $\mu$ is the dynamic viscosity, $F_1$ is the first blending function, and $F_2$ is the second blending function. The values of these constants are as follows [35]:

$\beta^* = 0.09$, $\alpha_1 = 5/9$, $\beta_1 = 3/40$, $\sigma_{k1} = 0.85$. $\sigma_{\omega 1} = 0.5$, $\alpha_2 = 0.44$, $\beta_2 = 0.0828$, $\sigma_{k2} = 1$, $\sigma_{\omega 2} = 0.856$.

The discrete phase method and the Euler–Lagrangian method were used to solve Equation (5) for particle motion [36]:

$$\frac{dV_P}{dt} = F_D(U - V_P) + \frac{g(\rho_P - \rho)}{\rho_P} + F_{others} \tag{5}$$

In the above formula,

$$F_D = \frac{18\mu C_D \text{Re}}{\rho_p d_p^2 \cdot 24} \tag{6}$$

where $F_D$ is the drag force per unit mass of the particle; $U$ is fluid velocity; $V_P$ is velocity; $\rho_P$ is particle density; $g$ is gravitational acceleration; $d_p$ is particle diameter; $C_D$ is drag coefficient; $F_{others}$ is other forces per unit mass, and $R_e$ is relative Reynolds number.

Erosion wear refers to the phenomenon that the solid particles or the fluid are corroded, worn, etc. caused by the surface of the material. The wear rate is usually used to measure the amount of wear of the material. The expression of wear rate is Equation (7) [37]:

$$R_{erosion} = \sum_{p=1}^{N_{particles}} \frac{m_p C(d_p) f(\alpha) v^{b(v)}}{A_{face}} \tag{7}$$

where $m_p$ is particle quality; $C(d_p)$ is particle diameter function; $\alpha$ is particle impact material surface angle; $f(\alpha)$ is impact angle function; $v$ is particle impact relative speed; $b(v)$ is relative speed function;

$A_{face}$ is particles on the surface of the material erosion wear area; and $N_{particles}$ is the number of particles hitting the material at the wear. The unit of $R_{erosion}$ is $kg/(m^2/s)$.

## 2.2. CFD Setup

Nowadays, blades are constructed mostly from composite structures; the outer layer is wrapped with alloy, and the inner layer is made from a composite. Ti-4Al-1.5Mn, Mg-Li9-A3-Zn3, and Al7075-T6 are commonly used aviation materials, and are suitable for simulation experiments with different physical properties [38–40], the density, hardness and compositional elements of these three materials are listed in Table 1. Among the prices of the three materials, magnesium alloys are the most expensive, while aluminum alloys are the least expensive. Generally, magnesium alloys are used for weight reduction because of their light weight; titanium alloys are used under high-strength conditions; aluminum alloys are mainly used to reduce costs as they are cheap and easy to manufacture. For this reason, these materials in the helicopter blade models were used in this study. The cross-sectional profile of the helicopter rotor blade, and the geometric details, are shown in Figure 1 for the classic airfoil NACA0012. Here, the authors emphasize the blade tip, as this area of the blade experiences the highest linear velocity and can easily become damaged [41].

**Table 1.** Elemental composition (%) and characteristics of three materials.

| | Ti | Al | Mn | O | Others | Density (kg/m³) | Hardness (Vickers) |
|---|---|---|---|---|---|---|---|
| Ti-4Al-1.5Mn | margin | 3.5–5.0 | 0.8–2.0 | 0.15 | 0.55 | 4500 | 330 |
| | Mg | Li | Al | Zn | others | | |
| Mg-Li9-A3-Zn3 | margin | 8.5–9.5 | 2.5–3.5 | 2.5–3.5 | 0.5 | 1510 | 158 |
| | Al | Zn | Mg | Cu | others | | |
| Al7075-T6 | margin | 5.1–6.1 | 2.1–2.9 | 1.2–2.0 | 1.5 | 2800 | 89 |

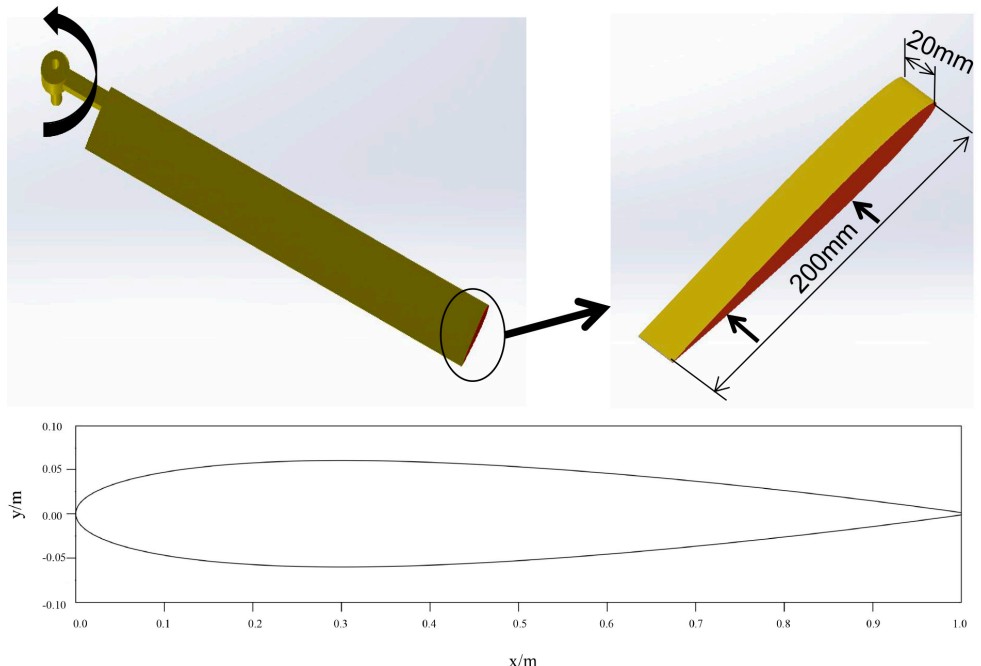

**Figure 1.** Geometric details of the helicopter rotor blade.

The particle type is set to silica crystal and its density is 2200 kg/m³. The Mohs hardness is 7. The flow rate of the particles was set to 0.97 kg/s. Particles with a regular 100-μm diameter are uniformly ejected from the inlet surface (size: 200 × 200 mm²) at different impact speeds (70, 150, or 220 m/s) and angles of attack (6°, 3°, or 0°). The jet velocity at the particle inlet surface was set to the impact velocity.

The boundary condition of the rotor blade is defined as the steady state. The simulation materials and parameters are shown in Table 2.

**Table 2.** Materials and parameters.

| Materials | Ti-4Al-1.5Mn | Mg-Li9-A3-Zn3 | Al7075-T6 |
|---|---|---|---|
| Sand Size (μm) | 100 | 100 | 100 |
| Impact Velocity (m/s) | 70 | 150 | 220 |
| Total Flow Rate (kg/s) | 0.97 | 0.97 | 0.97 |
| Angles of Attack (°) | 6 | 3 | 0 |

Figure 2a,b show the boundary conditions and mesh structure, respectively. The surface near the rotor blade was processed with mesh refinement, as shown in Figure 2b. In addition, the mesh independence of the erosion shield was studied by increasing the number of elements. There are 2,154,080 elements in the mesh structure in Figure 2b. The flow plant domain in Figure 2b passes the fluent grid quality test [42]. Table 3 is a grid independence test by increasing the number of elements. The average erosion rate and $|\Delta E_{Avg.}|$ are the criteria for detecting grid independence. The value of the average erosion rate is the result of Ti-4Al-1.5Mn being eroded by solid particles of 220 m/s, and $|\Delta E_{Avg.}|$ is the average erosion rate deviation. Through these two sets of values, the number of elements selected for this simulation is reasonable.

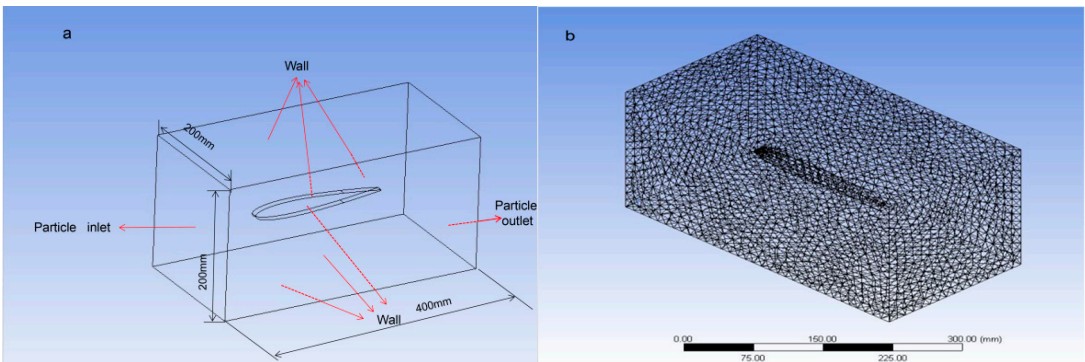

**Figure 2.** CFD model of the helicopter rotor blade: (**a**) boundary conditions; (**b**) mesh structure.

**Table 3.** Results of mesh independence study.

| $n$ | Number of Elements | Average Erosion [kg/(m²·s)] | $|\Delta E_{Avg.}|$ (%) |
|---|---|---|---|
| 1 | 1,498,140 | $1.73 \times 10^7$ | - |
| 2 | 2,154,080 | $1.68 \times 10^7$ | 2.89 |
| 3 | 4,943,170 | $1.59 \times 10^7$ | 5.36 |

After a preliminary blade erosion simulation, in this study, due to the fluent boundary condition setting, only the front end of the blade section was affected by the impact of the particles. As shown in Figure 3, the front end of the blade section is exposed to significant pressure. Therefore, when a bionic coating is added to the blade later, the bionic coating is applied on the front end of the blade section.

First, the erosion simulation was carried out on blades of the three materials under general sandstone conditions. The material with the best erosion resistance was determined after processing the simulation dates at three particle speeds and three impact angles.

The bionic structure of the back of a desert scorpion informed the construction of the bionic coating used by Han's team [43], as shown in Figures 4 and 5. V- and VC-type bionic coatings were examined in this study. The V-type biomimetic samples were assumed to be multiple standard trapezoids of the

same type, as shown in Figure 4. The VC-type bionic sample was based on the V-type sample with a convex hull, which was assumed to be regular hemispherical, as shown in Figure 5.

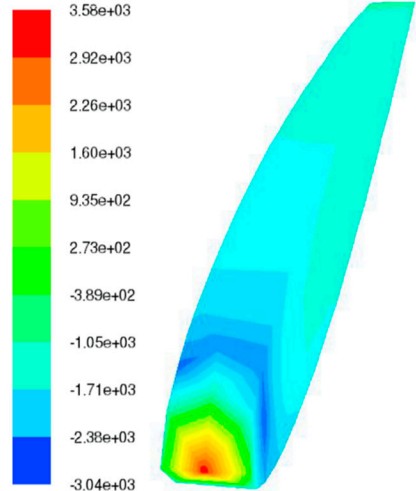

**Figure 3.** Model of blade pressure.

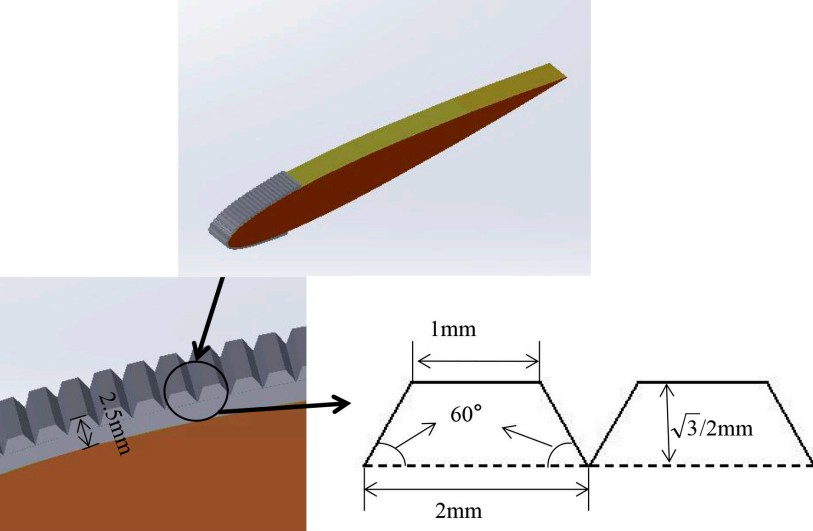

**Figure 4.** Bionic coating of V-type.

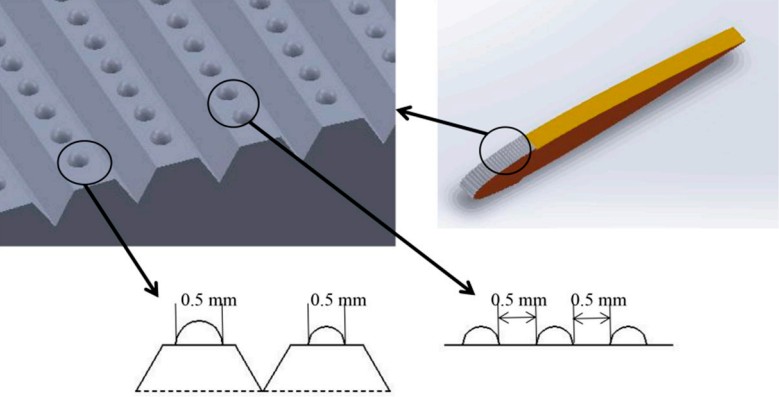

**Figure 5.** Bionic coating of VC-type.

The bionic coating on the leading edge of the blade was 2.5-mm thick. This model is meshed in the ANSYS mesh, and then the option parameters are imported into the fluent to be the same as the uncoated blade simulation. Erosion analysis was carried out at three impact speeds (70, 150, and 220 m/s) and three angles of attack (6°, 3°, and 0°). The simulation parameters are shown in Table 4.

**Table 4.** Materials and parameters with bionic coatings.

| Materials | Ti-4Al-1.5Mn | | |
|---|---|---|---|
| Type | V-type | | VC-type |
| Impact Velocity (m/s) | 70 | 150 | 220 |
| Total Flow Rate (kg/s) | 0.97 | 0.97 | 0.97 |
| Angles of Attack (°) | 6 | 3 | 0 |

## 3. Results and Discussion

### 3.1. Erosion Simulations with Different Impact Speeds

The focus of this paper is to study the erosion strength of the blades. Blade erosion or deformation can cause serious efficiency loss and vibration effects of the helicopter. The material of the blade itself is an important factor to deal with erosion. By studying the erosion of different particle impact speeds on different material blades, we look for the best erosion resistant material. Figure 6 shows the average area erosion ratios of Ti-4Al-1.5Mn, Mg-Li9-A3-Zn3, and Al7075-T6 at impact speeds of 70, 150, and 220 m/s. The erosion rate of the blade surface increased sharply with the impact speed, as shown in Figure 6. The erosion ratio in Figure 6 is defined as the volume fraction of material loss per kilogram of particles incident on the surface of the material. The values of erosion ratio are shown in Table 5. Thus, Ti-4Al-1.5Mn demonstrated superior anti-erosion performance at all impact speeds.

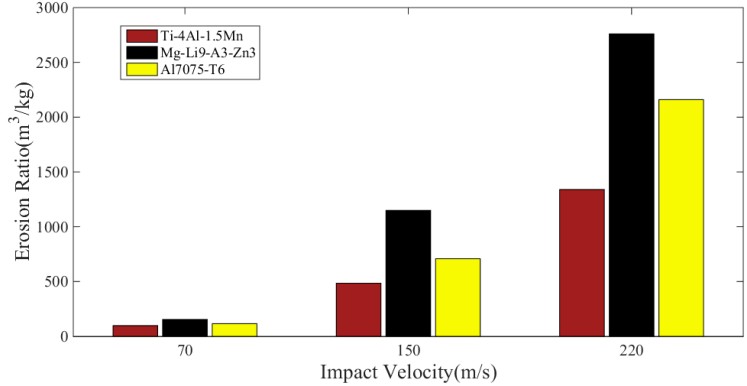

**Figure 6.** Erosion ratios of three materials at different impact speeds.

**Table 5.** Erosion ratios of three materials at different impact speeds.

| | Value (m³/kg) | | |
|---|---|---|---|
| | 70 m/s | 150 m/s | 220 m/s |
| Ti-4Al-1.5Mn | 98 | 485 | 1340 |
| Mg-Li9-A3-Zn3 | 154 | 1149 | 2760 |
| Al7075-T6 | 116 | 708 | 2110 |

Figure 7 shows the 3D accretion rate distribution on the surface of the rotor blades, when the attack angle was 0° at impact speeds of 70, 150, and 220 m/s. The accretion rate here is defined as the mass per square meter of accretion particles per unit time and is usually used to measure wear, the higher the accretion rate of the material, the more resistant to wear. Figures 8 and 9 show the accretion rate curve and the 3D erosion rate distribution for the helicopter rotor blade surface, the erosion rate

here is defined as the mass loss per square meter of material per unit time; the average accretion rates at different impact speeds are listed in Table 6. The erosion performance of the three materials differed significantly depending on the impact speed. Mg-Li9-A3-Zn3 showed the lowest performance, independent of the impact speed. The accretion rate of the Ti-4Al-1.5Mn material was much higher than those of the other two materials. The erosion of all three materials was most pronounced at the highest impact speed of 220 m/s. Thus, Ti-4Al-1.5Mn demonstrated the best erosion resistance in the three materials.

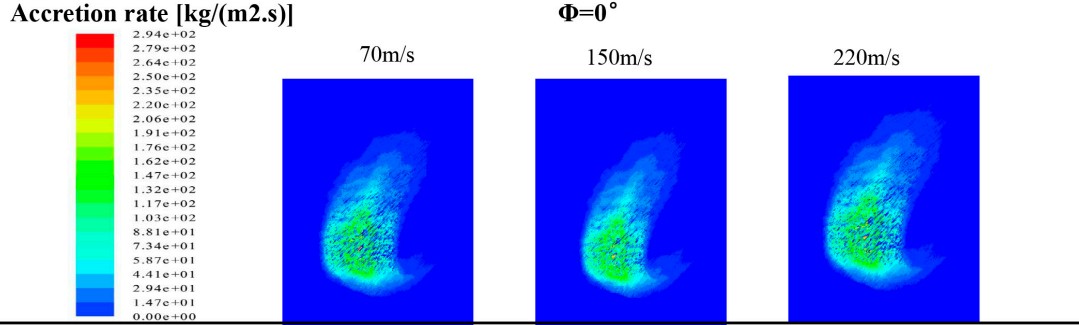

**Figure 7.** Three-dimensional accretion rates at different impact speeds at 0° angle of attack.

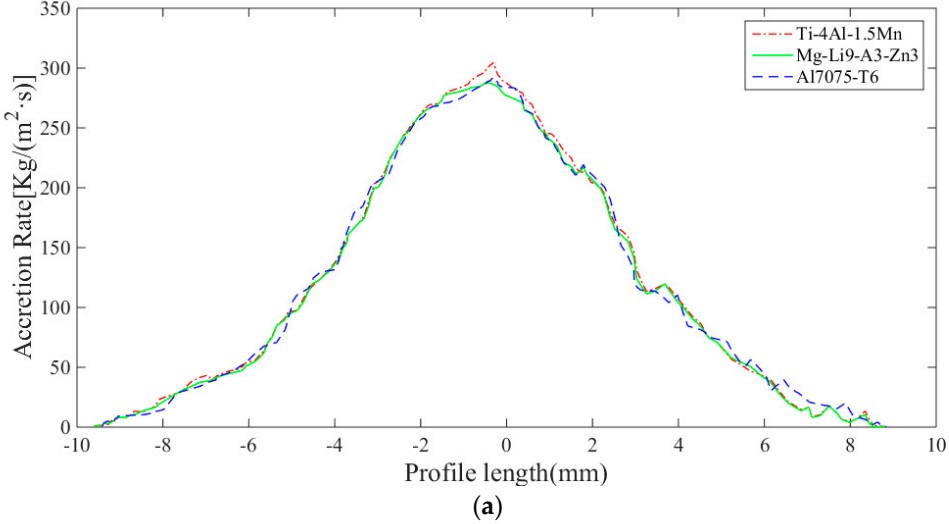

(**a**)

**Figure 8.** *Cont.*

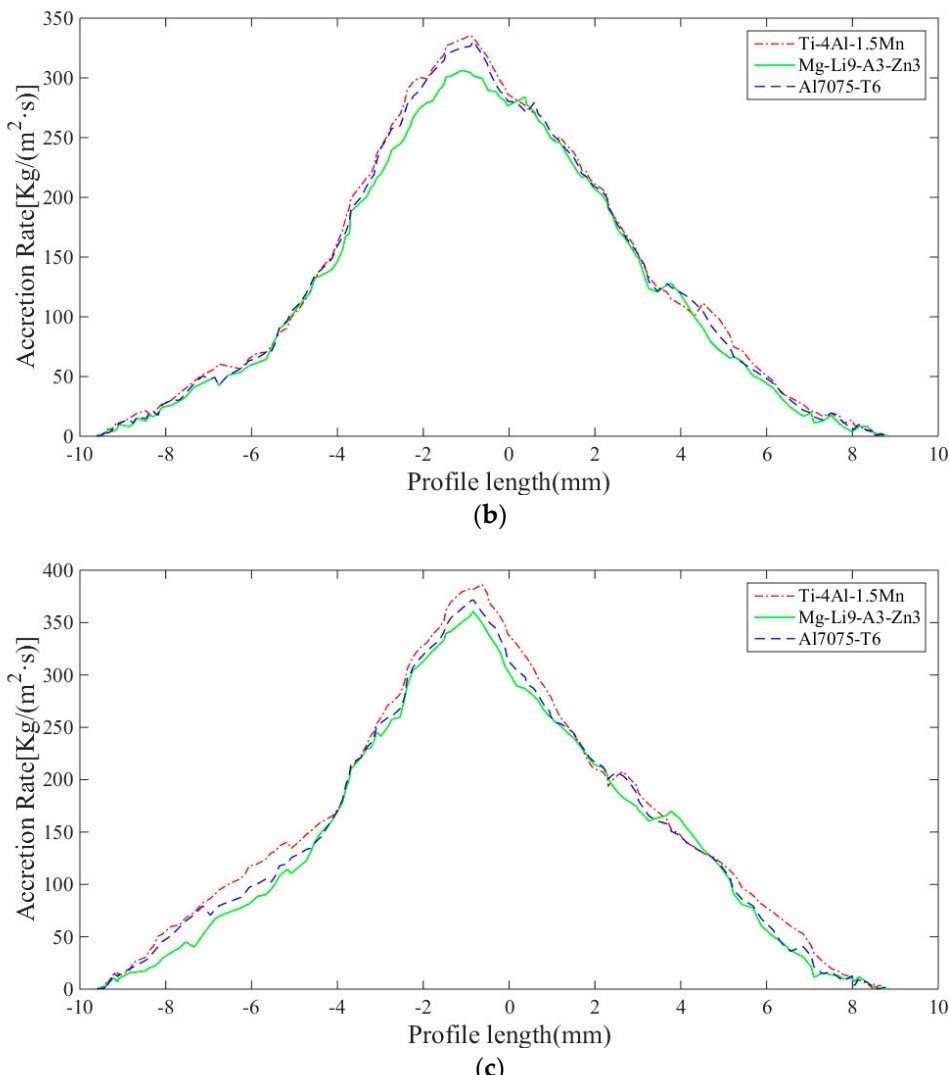

**Figure 8.** Accretion rate curves of three materials at different impact speeds at 0° angle of attack: (**a**) $V_P$ = 70 m/s; (**b**) $V_P$ = 150 m/s; (**c**) $V_P$ = 220 m/s.

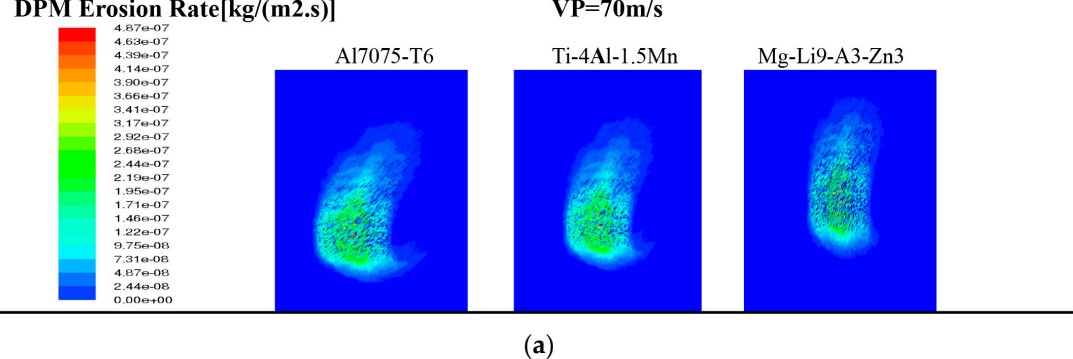

(**a**)

**Figure 9.** *Cont.*

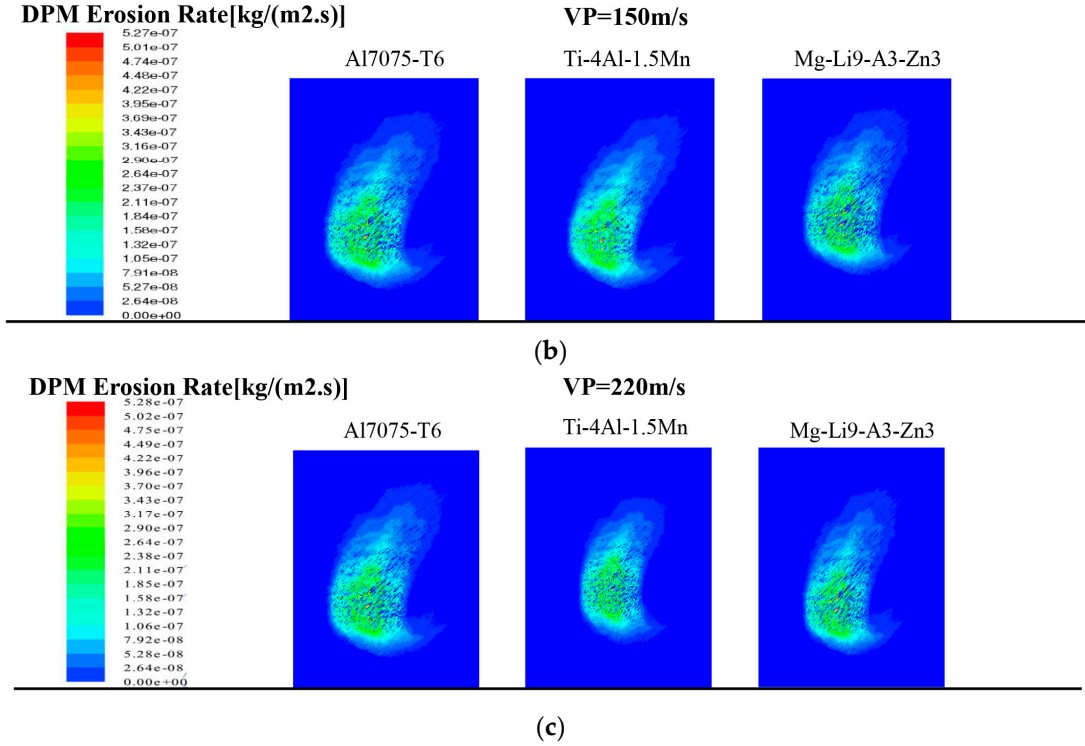

**Figure 9.** Three-dimensional erosion rates of three materials at different impact speeds at 0° angle of attack: (**a**) $V_P$ = 70 m/s; (**b**) $V_P$ = 150 m/s; (**c**) $V_P$ = 220 m/s.

**Table 6.** Average accretion rates of the three material at different impact speeds.

|  | Value [kg/(m$^2$·s)] | | |
|---|---|---|---|
|  | 70 m/s | 150 m/s | 220 m/s |
| Ti-4Al-1.5Mn | 124.85 | 138.86 | 162.46 |
| Mg-Li9-A3-Zn3 | 122.62 | 130.41 | 148.56 |
| Al7075-T6 | 124.84 | 136.05 | 154.23 |

### 3.2. Erosion Simulations at Different Attack Angles

Figure 10 shows the average erosion ratio of rotor blades at different attack angles for an impact speed of 220 m/s; Mg-Li9-A3-Zn3 exhibited more pronounced body erosion compared with the other two materials. Figure 11 displays the 3D erosion rate distribution over the erosion shield, and Figure 12 shows the erosion rate curve of the blade surface at an impact speed of 220 m/s for the different materials. The average erosion rate at different angles of attack was calculated. The results are listed in Table 7. The variance in the average erosion rate of Ti-4Al-1.5Mn, Mg-Li9-A3-Zn3, and Al7075-T6 was 0.0117%, 0.0116%, and 0.0167%, respectively.

The authors' results showed no obvious variation in the erosion rate with change in the attack angle; from this, it can be inferred that the angle of attack only affects the erosion area, which increases with the angle of attack.

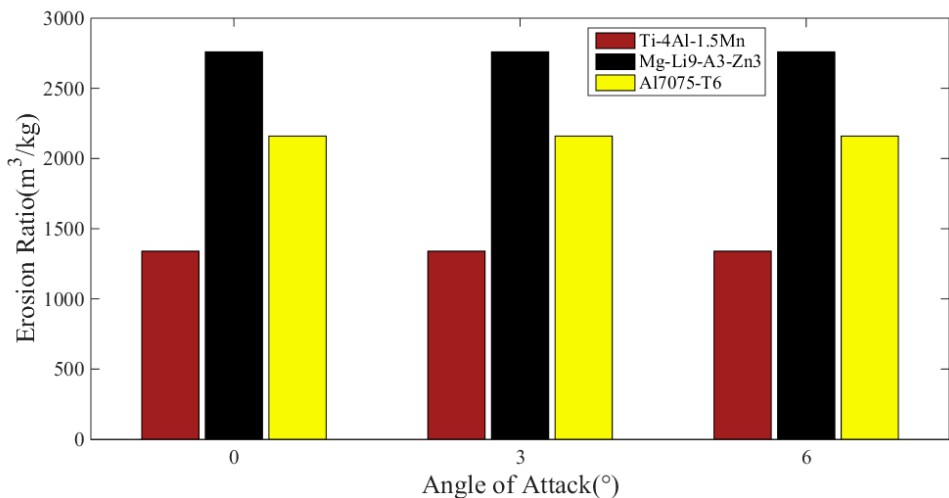

**Figure 10.** Erosion ratios of three materials at different angles of attack.

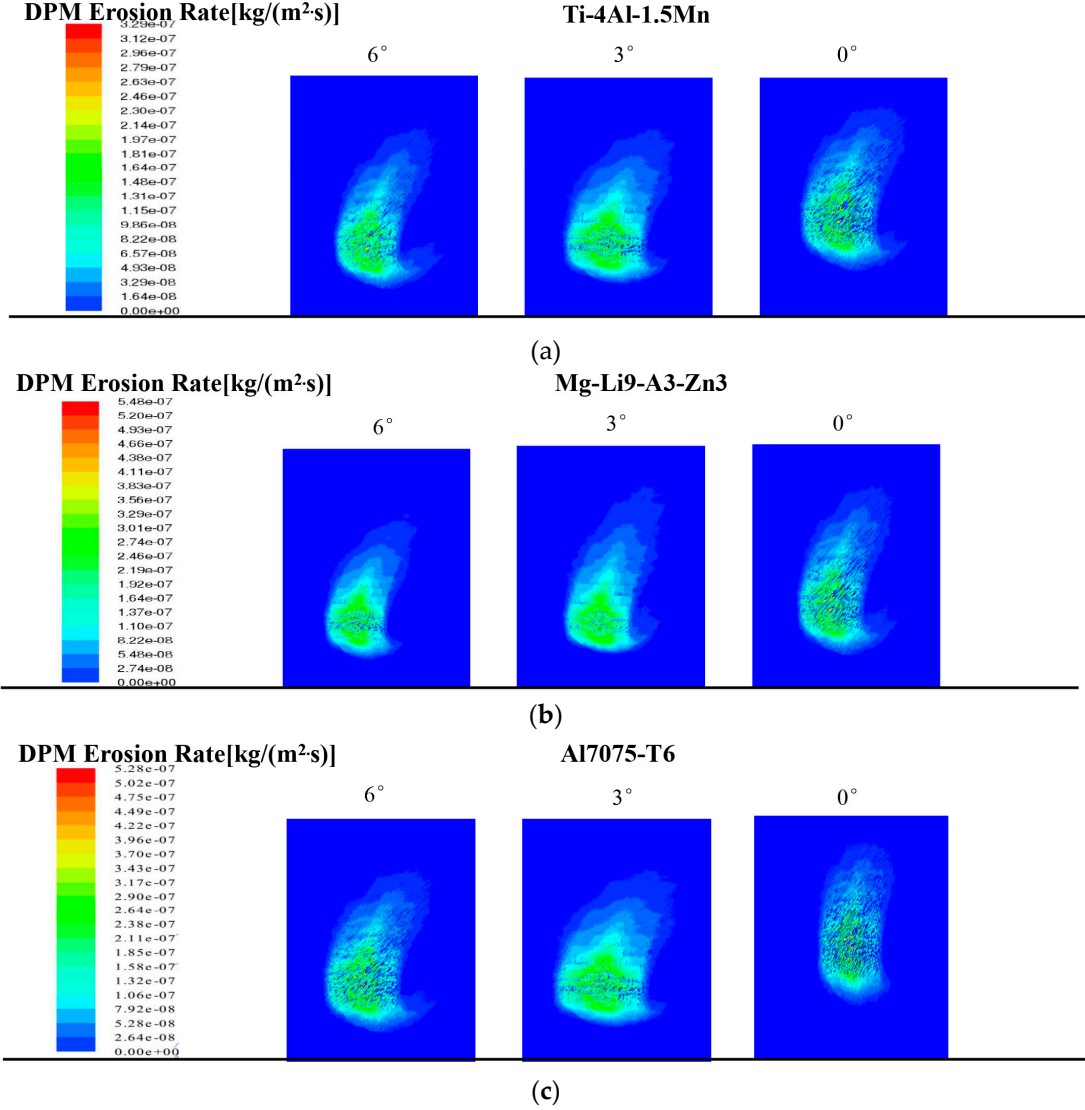

**Figure 11.** Three-dimensional erosion rates of the three materials at different angles of attack at impact speed of 220 m/s: (**a**) Ti-4Al-1.5Mn; (**b**) Mg-Li9-A3-Zn3; (**c**) Al7075-T6.

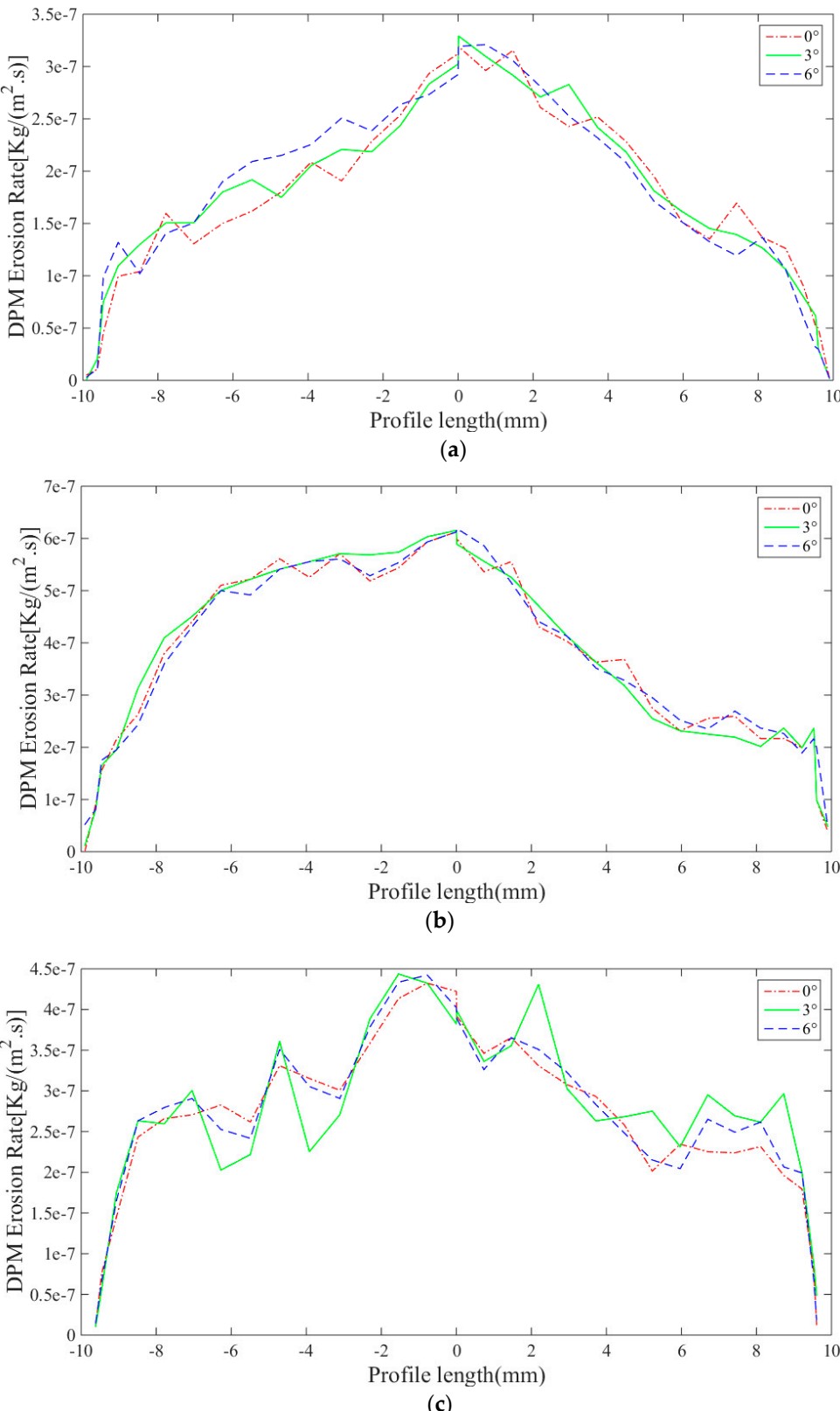

**Figure 12.** Erosion rate curves of the three materials at different angles of attack at impact speed of 220 m/s: (**a**) Ti-4Al-1.5Mn; (**b**) Mg-Li9-A3-Zn3; (**c**) Al7075-T6.

**Table 7.** Average erosion rates of the three material at different angles of attack.

| | Value [kg/(m²·s)] | | |
|---|---|---|---|
| | 6° | 3° | 0° |
| Ti-4Al-1.5Mn | $1.71 \times 10^7$ | $1.70 \times 10^7$ | $1.68 \times 10^7$ |
| Mg-Li9-A3-Zn3 | $3.60 \times 10^7$ | $3.60 \times 10^7$ | $3.57 \times 10^7$ |
| Al7075-T6 | $2.62 \times 10^7$ | $2.68 \times 10^7$ | $2.58 \times 10^7$ |

### 3.3. Erosion Simulations with Different Bionic Coatings

Based on the results presented in Sections 3.1 and 3.2, V-type and VC-type bionic coatings were applied to the outer layer of the blade covered by Ti-4Al-1.5Mn. Figures 13 and 14 show the 3D erosion rate and the erosion rate curve, respectively, for the two types of bionic coatings, given an angle of attack of 0°. The average erosion rate of the bionic coatings at different impact speeds are listed in Table 8. The V-type bionic coating exhibited better erosion resistance, with erosion rates 26.1%, 26.6%, and 27.4% that of the VC-type at the three impact speeds of 70, 150, and 220 m/s, respectively.

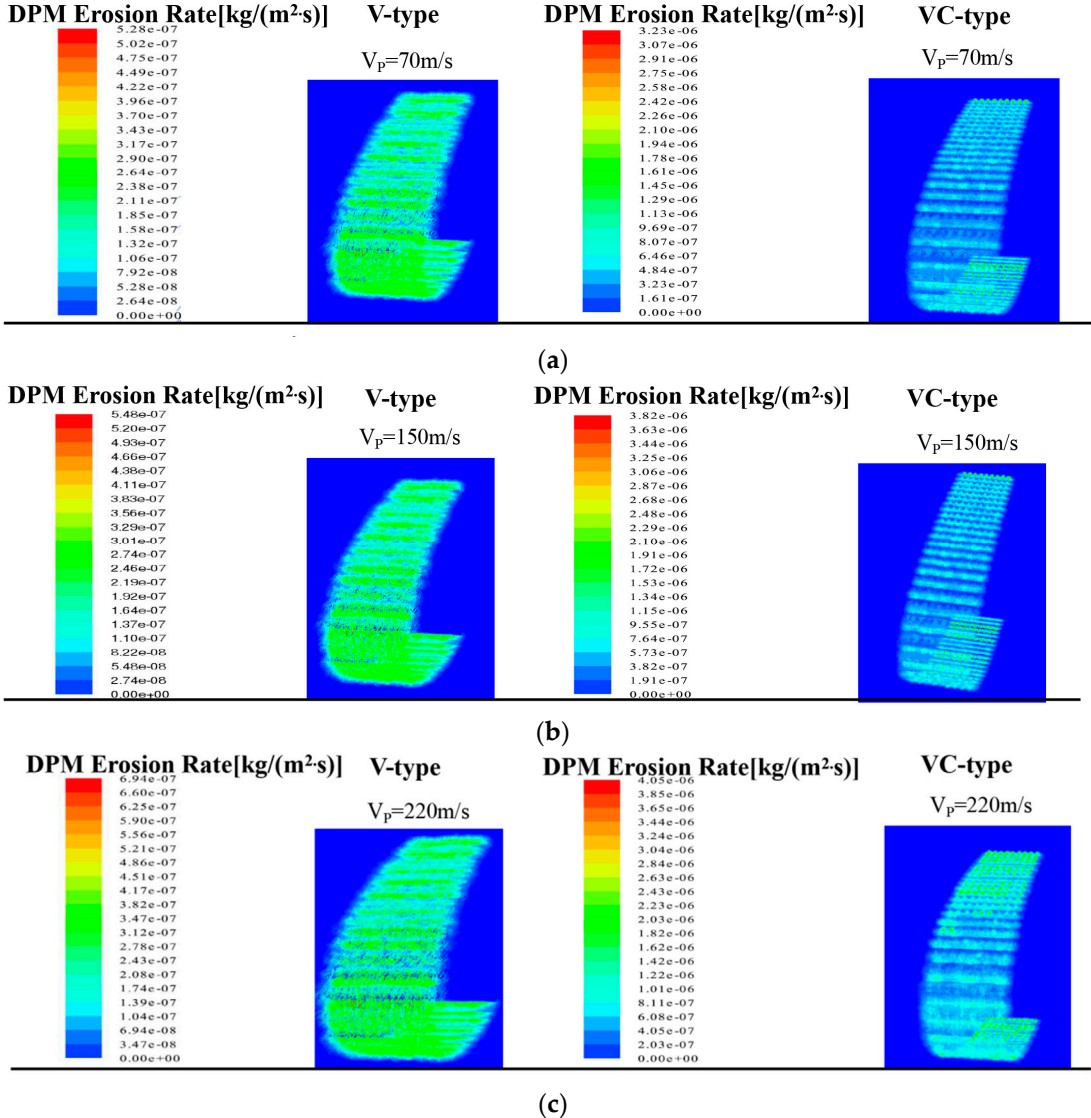

**Figure 13.** Three-dimensional erosion rates of the two bionic coatings at 0° angle of attack: (**a**) $V_P = 70$ m/s; (**b**) $V_P = 150$ m/s; (**c**) $V_P = 220$ m/s.

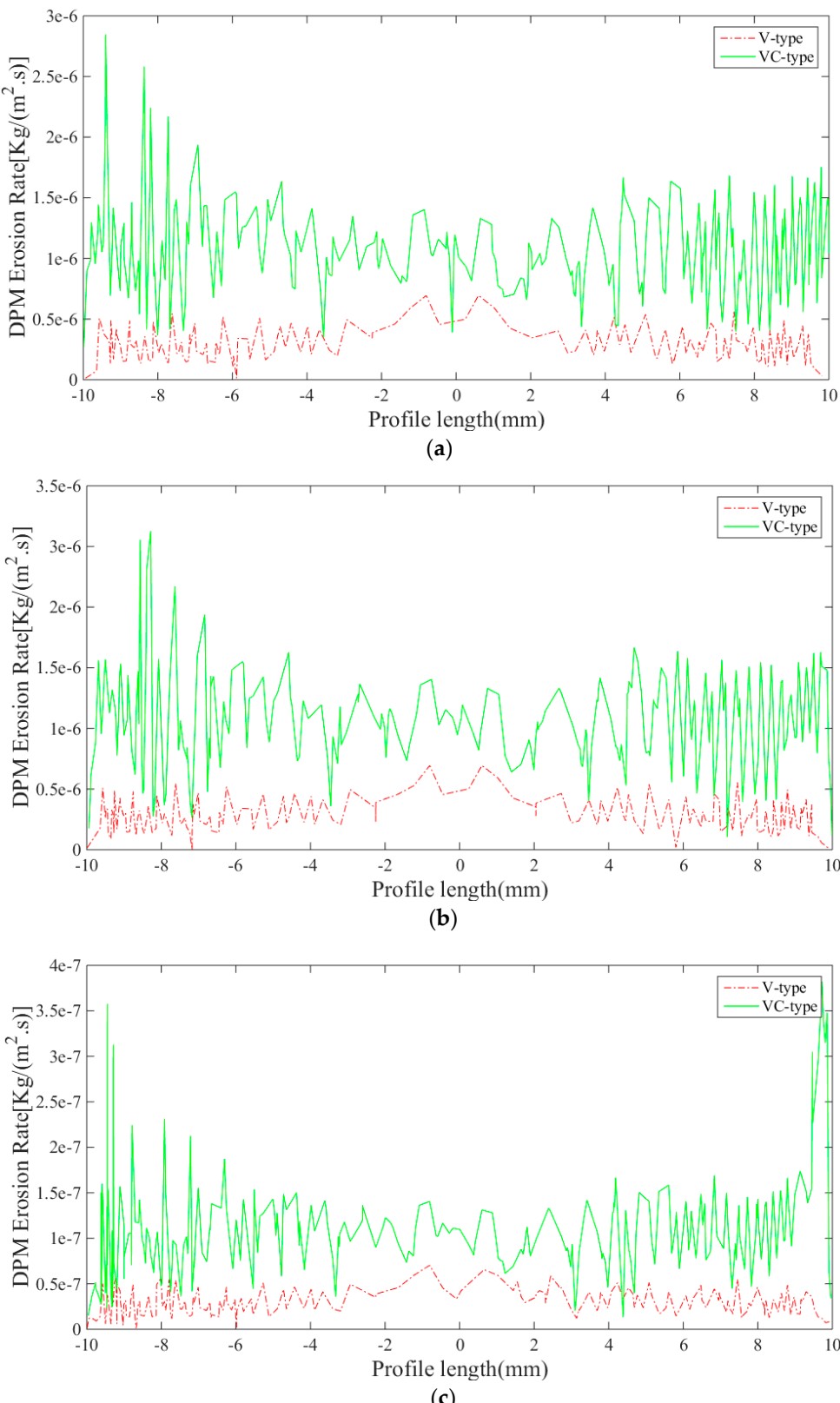

**Figure 14.** Erosion rate curves of bionic coatings at different speeds at 0° angle of attack: (**a**) $V_P$ = 70m/s; (**b**) $V_P$ = 150 m/s; (**c**) $V_P$ = 200 m/s.

**Table 8.** Average erosion rates of the two types of bionic coatings at different impact speeds.

| | Value [kg/(m²·s)] | | |
|---|---|---|---|
| | 70 m/s | 150 m/s | 220 m/s |
| V-type | $0.28 \times 10^6$ | $0.29 \times 10^6$ | $0.31 \times 10^6$ |
| VC-type | $1.07 \times 10^6$ | $1.09 \times 10^6$ | $1.13 \times 10^6$ |

Figure 15 shows the 3D erosion rate of the VC-type bionic coating at different angles of attack. Taken together, these results indicate that the erosion rate of the blades is almost independent of the angle of attack, even with the addition of a bionic coating.

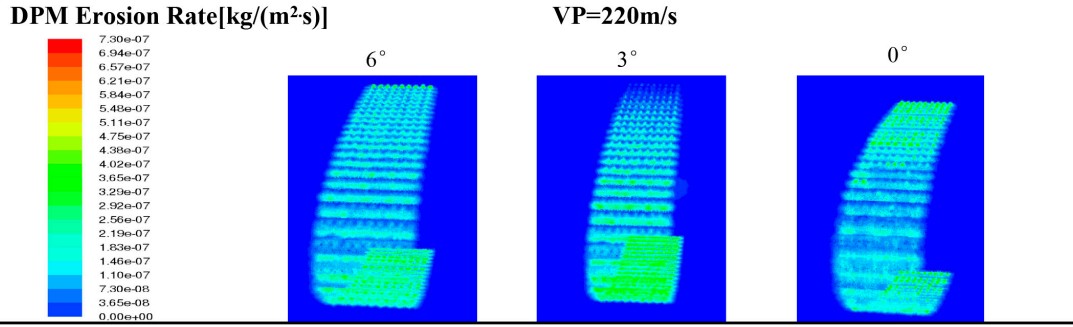

**Figure 15.** Three-dimensional erosion rates of bionic coating of V-type at impact speed of 220 m/s.

## 4. Conclusions

The erosion behaviors caused by solid particles impinging on helicopter rotor blades covered in Ti-4Al-1.5Mn, Mg-Li9-A3-Zn3, and Al7075-T6 coating materials, at different particle impact velocities and angles of attack, were numerically analyzed. In addition, two bionic coatings were applied to the blade leading edge. Ti-4Al-1.5Mn with the V-type bionic coating provided the best erosion resistance in the simulation experiments. The conclusions from our study are given below.

(a) In the simulations with different impact speeds, the average accretion rate of Mg-Li9-A3-Zn3 was 96.32% that of Al7075-T6, and the average accretion rate of Al7075-T6 was 94.99% that of Ti-4Al-1.5Mn. Ti-4Al-1.5Mn demonstrated the best erosion resistance, followed by Al7075-T6 and Mg-Li9-A3-Zn3. The erosion rate increased significantly with the particle impact velocity.

(b) In the simulations of different angles of attack, the variance in average erosion rate of Ti-4Al-1.5Mn, Mg-Li9-A3-Zn3, and Al7075-T6 was 0.0117%, 0.0116%, and 0.0167%, respectively. Due to the variance, the erosion rates of the three materials remained nearly unchanged with changes in the angle of attack. Based on this, we speculate that changes in the angle of attack may only affect the erosion area of the blade.

(c) In the simulation experiments of the bionic coating, the average erosion rate of the V-type bionic coating was about 26.1%, 26.6%, and 27.4% that of the VC-type coating at impact speeds of 70, 150, and 220 m/s, respectively. The erosion resistance of the V-type coating is significantly higher than that of the VC-type coating. The angle of attack had only a slight effect on the erosion rate of the two types of bionic coatings.

Previous researchers have only focused on the erosion of materials and have not done much research on how to prevent erosion. So, the research on the anti-erosion of helicopter blades using bionic coatings has been developed in this article. In future, the problems of 'What does solid blade erosion look like when a complete blade rotates?' and 'How does a bionic coating affect flying vibration?' should be discussed and studied.

**Author Contributions:** Conceptualization, Z.H. and J.Z.; methodology, X.B. and Y.Y.; software, X.B.; validation, Z.H. and S.Z.; investigation, S.Z.; resources, J.Z.; data curation, X.B.; writing—original draft preparation, X.B.;

writing—review and editing, Y.Y., Z.H. and J.Z.; supervision, Y.Y.; project administration, S.Z. and J.Z.; funding acquisition, Y.Y. All authors have read and agreed to the published version of the manuscript.

**Funding:** This research was funded by the National Key Research and Development Program of China, grant number 2018YFA0703300, Joint Fund of Ministry of Education for Equipment Pre-research, grant number 6141A02022131, the National Key R&D Program of China, grant number 2017YFC0602000, and Science and Technology Development Project of Jilin Province, grant number 20190303061SF.

**Conflicts of Interest:** The authors declare no conflict of interest. The funders had no role in the design of the study; in the collection, analyses, or interpretation of data; in the writing of the manuscript; or in the decision to publish the results.

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
