# Peer review of "Study of Solid Particle Erosion on Helicopter Rotor Blades Surfaces"

_applsci, doi:10.3390/app10030977_

Round 1
Reviewer 1 Report
Dear Authors,
Thank you for the impressive contribution. I must emphasise the importance of the task you presented for the aerospace industry today.
However, I would have several remarks, which, in my opinion, may improve your presentation:
Please, do not use personal pronouns in the text. Usually in the scientific paper, one shall introduce himself as "the Author(s)". The introductory part must be enhanced. You did the literature analysis very briefly. Apart from mentioning the references only, please provide the readers with comments and critical analysis of previous work done. Highlight drawbacks and missing elements. Please place your research clearly among the other researchers. Following the remark in 2, analysis and conclusion part should also bring the comparative study of your results and those recalled in the introduction. Sentences like: "Additional experimental testing is required for further design optimization" are not sufficient and look superficial. Please, provide more in-depth comments on your findings. Figure 3 is unreadable at first glance. If you can think about an alternative presentation. The model: I dare to say, that the model is not proper. If you browse the results, especially those for the smooth skin, we can observe that the distribution of accretion rates is roughly circularly symmetrical. I would expect the line of symmetry close to the place of stagnation instead. I would suggest the model of infinite span. What is the influence of the bionic skin on the laminar flow and overall drag, as well as the vibrations?Looking forward to your comments
Your reviewer
Author Response
Response to Reviewer 1 Comments
Dear reviewers,
Thank you for your valuable comments regarding our manuscript. We carefully revised our article following your suggestions, and our point-to-point answers can be found below. We hope that our manuscript now fulfills the high standards of Applied sciences.
Point 1: Please, do not use personal pronouns in the text. Usually in the scientific paper, one shall introduce himself as "the Author(s)".
Response 1: Thanks for your suggestion, personal pronouns have been replaced. (see L18,
L34, L47, L138)
Point 2: The introductory part must be enhanced. You did the literature analysis very briefly. Apart from mentioning the references only, please provide the readers with comments and critical analysis of previous work done. Highlight drawbacks and missing elements. Please place your research clearly among the other researchers.
Response 2: Thank you for your comments and opinions. Reviews of the current literature and descriptions of this paper have been added. (see L38-43, L49-51, L56-57, L63, L69-71)
Point 3: Following the remark in 2, analysis and conclusion part should also bring the comparative study of your results and those recalled in the introduction. Sentences like: "Additional experimental testing is required for further design optimization" are not sufficient and look superficial. Please, provide more in-depth comments on your findings.
Response 3: Your evaluation is accurate, the discussion of future research work have been added. The problem that ‘What does solid blade erosion look like when a complete blade rotates? How does a bionic coating affect flying vibration?’ have not been known clearly, all need to be discussed and studied. (see L304-308)
Point 4: Figure 3 is unreadable at first glance. If you can think about an alternative presentation.
Response 4: Thanks for your comment, Figure 3 has been replaced.
Point 5: The model: I dare to say, that the model is not proper.
Response 5: Your opinion is helpful to us, the model has been modified to ‘Erosion equations and CFD setup’. (see L90)
Point 6: If you browse the results, especially those for the smooth skin, we can observe that the distribution of accretion rates is roughly circularly symmetrical. I would expect the line of symmetry close to the place of stagnation instead. I would suggest the model of infinite span.
Response 6: Your analysis is accurate. We also noticed this phenomenon during the simulation. Considering the randomness of the particle incidence and the parameter setting relationship, it is reasonable that the curve is not completely symmetrical and the symmetry line is only close to the stagnation point. And the infinite span model and full blade simulation would be the focus of next research.
Point 7: What is the influence of the bionic skin on the laminar flow and overall drag, as well as the vibrations?
Response 7: Your comment is good. The bionic skin can achieve anti-erosion on blades, meanwhile it can also effect the flow field, overall drag and vibration of the flight. Compared to blades without bionic skin, the characteristics of flow field, overall resistance and vibration on blades with bionic skin are interseting topics. Furthermore,we are working on the topics and some meaningful conclusions could be reached.

Reviewer 2 Report
There are some typos that should be easily corrected: EXAMPLE: line 96M “none space after . or ,” Authors consider 3 actual manufacturing materials. I recommend to deep in their mechanical characteristics (and price) to light where and when can be used one or other. Section 3: For better comprehension, I recommended to refer the obtained results to the best one, not Mg… to Al, al Al to Ti-…. Section 2: check all used variables and parameters are properly defined The mathematical model described, ¿is implemented yet in Fluent or has been developed by authors in the code? The model, only considers the fluid part and not the solid? In than case: how can estimate the erosion for different materials at different strain rate? If the model includes the solid part: Where are the constitutive model for the blade materials? How is estimated the strain rate and failure parameters? Expression 7: I recommend to specify the units. Where comes the expression 7? From literature? Where are defined the functions f(alpha), b, …etc? Where are included the particles material mechanical properties (density, hardness or other constitutive properties as the Young’s modulus, or yield stress…)? Section 2.2: Authors include elemental composition but not mechanical properties that are relevant in this problem and should be considered somewhere to estimate the erosion.Line 136 and table 3: I don’t understand the number format: XXX,XXXX? Line 141: I recommend “element or mesh size” better that number of grids. Which is the element nomenclature in Fluent? Could be improved the quality of Figure 3? It is not easy to see the blade geometry. Maybe it could be superposed. (similar to Fig. 7,9, 11, 13 and 15) How affect the V and VC-type to the element mesh size? Please, check the results in lines 178-179 (check the position of “,” and the number of following significative characters). Maybe is better to put results in a Table. If erosion rate is the impact angle is independent of attack angle, erosion area and erosion depth don’t (erosion volume is quite constant) , is it correct?
Author Response
Response to Reviewer 2 Comments
Dear reviewers,
Thank you for your valuable comments regarding our manuscript. We carefully revised our article following your suggestions, and our point-to-point answers can be found below. We hope that our manuscript now fulfills the high standards of Applied sciences.
Point 1: There are some typos that should be easily corrected: EXAMPLE: line 96M ‘none space after . or ,’.
Response 1: I am sorry for our negligence. We have made changes. We have checked the
rest of the article and corrected the error.
Point 2: Authors consider 3 actual manufacturing materials. I recommend to deep in their mechanical characteristics (and price) to light where and when can be used one or other.
Response 2: The composition and mechanical properties of the three materials have been listed in Table 1. Among the prices of the three materials, magnesium alloys are the most expensive, while aluminum alloys are the least expensive. Generally, magnesium alloys are used for weight reduction because of light weight; titanium alloys are used under high-strength conditions; and aluminum alloys are cheap and easy to manufacture, and are mainly used to reduce costs. (see L132-136)
Point 3: Section 3: For better comprehension, I recommended to refer the obtained results to the best one, not Mg… to Al, al Al to Ti-….
Response 3: Your opinion is very correct, we have directly changed the result obtained to the best one. (see L212-213)
Point 4: Section 2: check all used variables and parameters are properly defined The mathematical model described, ¿is implemented yet in Fluent or has been developed by authors in the code?
Response 4: We checked the variables and parameters used and made changes. (see L102-104)
These mathematical models have been implemented in Fluent.
Point 5: The model, only considers the fluid part and not the solid? In than case: how can estimate the erosion for different materials at different strain rate? If the model includes the solid part: Where are the constitutive models for the blade materials? How is estimated the strain rate and failure parameters?
Response 5: Your comments are very comprehensive and meaningful. The bottom layer models of software FLUENT have been designed with the problem. Then the bottom layer models have been processed and packaged to slove the problem. Therefore, the solid part and fluid part not specifically pointed out in this paper. Meanwhile, the defination of erosion ratio has no obvious relation with the strain rate and failure parameters. (see L122)
Point 6: Expression 7: I recommend to specify the units. Where comes the expression 7? From literature? Where are defined the functions f(alpha), b, …etc?
Response 6: The unit and reference [37] and reference [42] of expression 7 has been added. The definition f(alpha) and others are shown in the references.
Tilly, G. P. Erosion caused by impact of solid particles. on Mate. Sci. and Techn.1979, 13, 287-319.[42] Fluent. ANSYS fluent theory guide. Canonsburg, PA: ANSYS.2010.
Point 7: Where are included the particles material mechanical properties (density, hardness or other constitutive properties as the Young’s modulus, or yield stress…)?
Response 7: The solid particles are sandstone particles, the main component is SiO2, the density is 2200 kg/m3, and the Mohs hardness is 7. (see L143-144)
Young's modulus, yield stress and other constitutive properties are defined as default values in the software.
Point 8: Section 2.2: Authors include elemental composition but not mechanical properties that are relevant in this problem and should be considered somewhere to estimate the erosion.
Response 8: Your comment is helpful to us, the density and Vickers hardness of three materials have been added in Table 1.
Point 9: Line 136 and table 3: I don’t understand the number format: XXX,XXXX?
Response 9: We are sorry for this error, the formats have been changed to XXX,XXX. (see L162 and Table 3)
Point 10: Line 141: I recommend “element or mesh size” better that number of grids.
Response 10: Your suggestion is good and ‘number of grids’ has been changed to ‘element or mesh’. (see L156 and L159)
Point 11: Could be improved the quality of Figure 3? It is not easy to see the blade geometry. Maybe it could be superposed. (similar to Fig. 7,9, 11, 13 and 15) How affect the V and VC-type to the element mesh size?
Response 11: We are very sorry for this error. Figure 3 has been replaced completely. As the blades with V and VC coating are more complex than ordinary blade structures, the number of elements mesh have been increased in the process of intital simulation.
Point 12: Please, check the results in lines 178-179 (check the position of “,” and the number of following significative characters). Maybe is better to put results in a Table.
Response 12: Your suggestion is very useful, the values of erosion rate have been shown clearly in Table 5.
Point 13: If erosion rate is the impact angle is independent of attack angle, erosion area and erosion depth don’t (erosion volume is quite constant), is it correct?
Response 13: What you said is accurate. When the erosion conditions (impact velocity, particle size, etc) are constant, the erosion rate is relatively constant. If changing the angle of attack, there is no significant change in erosion rate, the erosion area and depth could be affected. (see L238-240)

Round 2
Reviewer 1 Report
Dear Authors,
Thank you for updating the paper.
Best regards
Your reviewer
Reviewer 2 Report
Thank you for clarify suggestions for a better comprehension.